# Impacts of Job Standardisation on Restaurant Frontline Employees: Mediating Effect of Emotional Labour

**Omar Chehab** [1] , **Shiva Ilkhanizadeh** [2,*] and **Mona Bouzari** [3]

1   Department of Business Administration, Cyprus International University, Lefkosa 99510, Turkey; omarchehab@outlook.com
2   School of Tourism and Hotel Management, Cyprus International University, Lefkosa 99510, Turkey
3   School of Tourism and Hotel Management, European University of Lefke, Lefke 99728, Turkey; mbouzari@eul.edu.tr
*   Correspondence: silkhanizadeh@ciu.edu.tr

**Abstract:** Managers of food service operations standardise various aspects of operations to sustain consistent service quality. Frontline employees in these operations are expected to carry out tasks as per standards. Standards demand that frontline employees regulate their behaviours and emotions to complete their duties. Therefore, referring to the organisational role theory and the emotion regulation theory as the directing basis, this study examined the impact of job standardisation on emotional labour, as well as the effect of emotional labour on emotional exhaustion and job satisfaction of frontline employees in the hospitality sector. This study also examined the mediating effect of emotional labour on the relation between job standardisation, on one hand, and emotional exhaustion and job satisfaction on the other hand. The data collection was carried out in food service operations in Lebanon. Structural equation modelling (SEM) was used to assess the relations. The results showed that job standardisation negatively affected emotional labour and that emotional labour had a positive effect on emotional exhaustion and a negative effect on job satisfaction. Furthermore, emotional labour mediated the relation between job standardisation and emotional exhaustion and job satisfaction. Practical and theoretical implications and directions for future research are also provided.

**Keywords:** job standardisation; emotional labour; emotional exhaustion; job satisfaction; frontline employees; hospitality sector

## 1. Introduction

Tourism continues to be an important generator of employment opportunities worldwide. The World Travel and Tourism Council (WTTC) [1] reports that 1 in every 10 jobs created globally is in the tourism sector. However, the tourism and hospitality industry workforce market shows challenging aspects [2]. The industry is characterised by seasonal fluctuations, long working hours, changing work shifts [3], and demanding customer experiences [4]. The hospitality sector is also known for a high degree of staff turnover, labour and skills shortages, a high proportion of seasonal, part-time, and on-call workers, in addition to a high proportion of students employed [2]. Consequently, managing human resources in the hospitality industry is important due to its impact on the service level as perceived by customers [2,5].

The inherently intangible and diverse essence of the hospitality service makes it exceedingly difficult to ensure service quality ahead of time [6]. Hospitality services take place in a social services environment [7] where strong and regular social encounters take place [8]. The service quality sometimes relies on factors such as the smile of servers, therefore guaranteeing that staff are productive and dedicated to their companies will lead either to success or failure [5]. Even more, hospitality provides an intangible service in

relevance to the providing employee [5]. The quality and meaning of a service varies from one service provider to another, from client to another, and from day to day [9].

Due to the increasing competition and shortage of competent employees [10], and to the interactive and intangible nature of the hospitality industry, hospitality businesses aim to establish uniformity by requiring performers to behave in a similar way through standardisation [11].

Standardisation is defined as "an agreed upon way of doing something" [12] (p. 1). Standardisation has been found as a key driver of increased international growth of service providers [13], which justifies the expansion of standardisation in the service industry over the last decades [14]. It has been empirically proven that standardisation of the work process positively affects both the firm and its customers [15]. Managers instruct their employees to follow a standardised operating process to achieve high service quality [16]. Standardisation clarifies procedures [17], improves processes [18], introduces employees to their work unit's visions and goals, and improves job satisfaction (JS) [19].

Standardisation has positive effects on employees as well. Hsieh and Hsieh [16] claim that standardisation clarifies the task's nature and content and reduces its complexity. They add that standardisation reduces role conflicts and clarifies the performance standards of businesses. Chang et al. [20] state that standardisation leads to greater speed and efficiency of task performance. Additionally, standardised jobs guarantee that service quality delivery is sustained by employees [9].

In hospitality settings, frontline employees' (FLEs) performance determines service organisations' degree of success [21]. In such a setting, an FLE is "the face of the first and often the only interaction between the service organisation and its customers" [22] (p. 369). FLEs play a significant role in customer satisfaction and expectations of service quality [23]. However, and because of FLEs' direct customer contact, it is argued that they are usually placed in an incredibly stressful setting [24]. FLEs face situations where they are required to respond to changing customer demands, satisfy the formal role demand, provide appropriate solutions to customer problems, and follow management standards [25].

In standardised service operations, encounters between FLEs and guests are depicted as part of the service input process [26]. Process controls include different methods (e.g., operating procedures) that are aimed at manipulating employees' behaviours and responses during operations [27]. Pizam [28] states that hospitality staff members should be enthusiastic and friendly when serving customers, even if they are in a bad mood or are faced with difficult customers. Encounters in hospitality and tourism operations may be emotionally charged [29]; thus, operations require employees to apply several emotional display rules, such as "always smile at the customer" [30] (p. 38). Hochschild [31] (p. 7) labels this "management of feeling to produce a publicly visible facial and bodily display" as "emotional labour" (EL).

Research shows that EL has positive and negative associations with FLEs' job outcomes. EL causes increased employee self-efficacy and self-esteem [32], increased job engagement [33], organisational commitment [34], and customer satisfaction [35]. However, EL also induces emotional exhaustion [36] and causes job stress, which affects JS [37].

Maslach and Jackson [38] define emotional exhaustion (EE) as the feeling of being overextended and depleted of one's emotional capital by occupational responsibilities. Additionally, Maslach et al. [39] present EE as a person's inability to meet the emotional demands of a job over an extended duration. Xu et al. [40] suggest that identifying the EE context will enable hospitality managers to address the problem of why employees remain exhausted and complain about their jobs. Furthermore, the stressful settings that FLEs face in their jobs will adversely affect their JS, defined as employees' emotional state and affective responses to their jobs [41]. Research shows that job standardisation is one way of dealing with the exhibited excessive job demands and stress. Job standardisation is claimed to clarify work content, increase role certainty, reduce role discord, explain the

performance standards of a service firm, and improve service quality by providing a more consistent environment [16].

Referring to the role theory [42], Katz and Kahn [43] state that the assignment of job roles prescribes the actions that workers are supposed to perform in order to complete their tasks and duties effectively. Furthermore, implying the emotion regulation theory [44,45], it is presumed that FLEs in restaurant operations regulate their emotions to adhere to the display rules and standards imposed and observed by managers. Emotion regulation is, in turn, expected to influence FLEs' EE and JS. Therefore, using the data collected from FLEs in Lebanon, this study aims to examine (1) the impact of standardisation on FLEs' EL, (2) EL's direct bearing on EE and JS, and (3) the mediating effect of EL on standardisation and employee EE and JS. These relations are explained by the role theory [42] and the emotion regulation theory [44,45].

This study aims to address several gaps in the literature and contribute to the understanding of the standardisation of FLE roles in services. Wakke et al. [46] report that although many economies are now dominated by services and despite the increasing standardisation, research about standardisation in services is relatively scarce. Tsaur et al. [9] also argue about the need for a deeper understanding of job standardisation and its effects. Since very few studies have tackled the effect of job standardisation on FLEs' job behaviours and outcomes, Reif et al. [47] have called for a deeper understanding of the acceptance and expression indicators of process standardisation. Furthermore, although EL is believed to have multifaceted effects on FLEs, McGinley et al. [48] (p. 492) state that "the research gap regarding EL in the hospitality industry is striking". Likewise, Wen et al. [49] have called for further studies on EL to gain a better understanding of its role in hospitality settings. The limited research (i.e., [50]) that examined the effects of job standardisation on FLEs' EL in a hospitality setting is remarkable. Notably, a review of EL literature reveals that its possible mediating effects have only been tested by a few researchers [49,51,52].

Briefly, this inquiry aims to expand the employment of the role theory and the emotion regulation theory in examining the relations among the various elements of the conceptual model in a hospitality setting. The outcomes of this inquiry are expected to contribute to the narrative of job standardisation in services and to increase the knowledge regarding EL and its effects on EE and JS. Examining EL as an outcome of standardisation and as a mediator will also add to the relatively limited research that has adopted EL in similar constructs. Finally, this study aims to provide relative implications for hospitality research and management.

As exhibited in the following sections, the remainder of the paper is structured as follows:Section 2 consists of the literature review, hypotheses development, and the conceptual model. The instrument for data collection and method of analysis are discussed in Section 3. Section 4 consists of a review of respondents' profiles and empirical findings. The discussion, theoretical, and practical implications of the study are presented in Section 5, while the limitations and directions for future research are stated in Section 6.

## 2. Literature Review and Hypotheses

This research attempts to combine the fundamental principles of the role theory and the emotion regulation theory as the basis for this study. The primary focus of the role theory [42] narrative is to explain what a job role is and how employees operate according to prescribed organisational roles, noted as "role expectations". According to Turner [53], "role" refers to a cluster of behaviours and attitudes that are assumed to belong together, such that a person is observed to act consistently when fulfilling the various elements of a single role and inadequately when failing to do so. Several researchers have used role theory concepts to study employees' organisational roles and respective behaviours (e.g., [54,55]).

Referring to the role theory and to fulfil role expectations, FLEs find themselves obliged to respond to varying customer requests, comply with the formal role obligations of the position, satisfactorily solve customer problems, and meet management expectations [25].

This process corresponds to the fundamental principles of the emotion regulation theory [44,45], which posits that employees are obliged to act according to the organisational display rules and standards imposed by the companies that they work for. Ekman [56] presents display rules as the standards of conduct that indicate not only which emotions are acceptable in each situation faced by employees at work but also how certain emotions should be communicated or expressed publicly.

Considering the role theory and the emotion regulation theory, this study suggests that hospitality employees in standardised service roles are expected to perform according to predefined standard operating procedures and to regulate their emotions in order to display those expected from them according to their positions and the various service dyads in which they are daily involved. The subsequent sections elaborate on the research constructs' direct and mediated relations.

### 2.1. Standardisation and EL

Job standardisation describes how FLEs abide by operating procedures to carry out their tasks [16]. According to Murase and Bojanic [57], food service operations are distinguished by a high degree of consistency and standardisation that covers aspects such as logo, menu, uniform, decoration, service, and food-preparing techniques. Hsieh and Hsieh [16] assert that a high degree of job standardisation will probably result in clear principles and techniques overseeing work tasks that workers should adhere to in order to achieve the service goals. This way, managers may influence their FLEs' responses to further improve the quality of service they provide [27], and employees in a wide assortment of service positions may recognisably act in conformance with some predefined service roles [58].

Standardisation sets a service operating protocol, which is provided to clients when they visit the organisation [20]. Standardisation is said to promote consistency [59], optimise service time [60], ensure good quality service to clients [61], encourage competition [18], reduce operating costs, maintain a brand image, and promote innovation in franchised operations [62]. Furthermore, standardisation reduces role ambiguity and clarifies the service firm's performance expectations [63], prevents inconsistencies in FLEs' behaviours when delivering services [64], and promotes the socialisation process for new workers easily and efficiently [65].

A literature review reveals that standardisation has several negative aspects as well. Hsieh and Hsieh [16] conclude that standardisation takes a significant number of operating decisions out of the hands of FLEs, which may lead to dissatisfaction. They add that standards hinder FLEs from using their know-how to solve customer problems. Vargo and Lusch [66] claim that job standardisation may damage FLEs' sense of service quality because it reduces the level of control exercised by FLEs.

Because of the extreme lack of studies that assess the effect of standardisation on employees' EL in hospitality settings, this research uses the constructs of the role theory and the emotion regulation theory to examine this relation. Given the scarce literature, this inquiry would enable scholars to gain better insights into the relation between standardisation and EL, specifically in hospitality settings.

EL refers to manipulating human emotions by counterfeiting, enhancing, or suppressing emotional expressions for the purpose of earning an income and/or achieving organisational goals [45]. EL is believed to have several positive effects in the work environment. According to Psilopanagioti et al. [67], EL can help employees display positive emotions in their work performance. Furthermore, EL has been found to be important in developing interpersonal listening, oral communication, and negotiating skills [68], and in facilitating interpersonal relationships as well [69]. Hur et al. [70] argue that paying attention to employees' EL helps them withstand stress in harshly competitive organisational environments, enhance their company loyalty, and share positive energy with colleagues to achieve goals. It is also reported that EL limits work stress [71].

However, long-term EL brings negative results for individuals [72]. In hospitality settings where interpersonal interactions form the core of service provision, Lee et al. [73] argue that emotional demands often lead to burnout, poor service performance, job dissatisfaction, and possibly turnover intention.

Pizam [28] argues that hospitality workers are acutely vulnerable to the principle of EL due to various service beliefs. They are forced to remain optimistic, polite, and smiling, often in situations that elicit adverse emotional responses to undesirable service experiences. In standardised food service operations, FLEs are expected to follow standardised rules and regulations of encounters and coordinated processes to complete their responsibilities and interact with clients [74]. They are therefore required to adhere to explicit display rules, policies, and procedures [16]. In the absence of standardised conduct, employees are able to adapt their behaviour when interacting with customers [75]. They are, therefore expected to engage in EL to a greater extent. Thus, the assumption of this research aligns with Morris and Feldman's [76] that the more psychological energy and physical commitment the service jobs demand from employees, the more emotional displays these jobs will involve. Henceforth, this study proposes that the degree to which FLEs in hospitality operations regulate their emotions is relative to the degree of standardised rules, policies, and regulations enforced in the workplace. To sum up, in this study, it is argued that increased levels of control via standardisation would negatively affect the EL mobilisation level of FLEs, formulated in this hypothesis:

**Hypothesis 1 (H1).** *Job standardisation is negatively related to employees' emotional labour.*

### 2.2. EL, EE, and JS

Previous research has proven the relations between employee EL and the number of outcomes at work. Based on the emotion regulation theory, Yin et al. [77] stress that as people manage their emotions, their bodies turn resources into energy to react to the current situation. This causes resource drain, which makes the nervous system react adversely to emotion regulation. In the long run, this resource drain would lead to EE. Additionally, Grandey [45] (p. 97) states that managing emotions will lead to emotional dissonance—"a state wherein the emotions expressed are discrepant from the emotions felt", which in turn will cause EE. Grandey [45,78] further notes that service providers experience greater levels of EE because their emotions are constantly either suppressed or exaggerated. Hakanen et al. [79] add that employees are emotionally exhausted when physical and emotional pressures drain their emotional energies. In their recent studies, Kwon et al. [80] and Back et al. [37] also conclude that prolonged situations demanding emotional regulation have a positive connection to employee EE.

Furthermore, most of the EL literature has focused on the connection between EL and employee satisfaction [81]. Relating to the emotion regulation theory, it is argued that while surface acting is said to have a negative impact on JS [82], deep acting is claimed to have a positive impact [83]. Grandey [45] concludes that in addition to EE, job dissatisfaction is suggested as another outcome of the dissonance dimension resulting from EL. This dissonance can make the person feel fake and hypocritical, which may eventually lead to personal and job dysfunction, with job dissatisfaction as one of its perceived outcomes [32].

Despite the immense acknowledgement and empirical research regarding the effects of EL in businesses, research leading to understanding its possible outcomes in the hospitality setting seems to be unrecognised [48]. Research has contemplated the detrimental effects of EL on EE and JS [37,71]. Accordingly, this study proposes that FLEs who engage in high levels of EL in the workplace are increasingly emotionally exhausted and dissatisfied with their jobs. Therefore, this paper puts forward the following hypotheses:

**Hypothesis 2a (H2a).** *Employees' emotional labour is positively related to emotional exhaustion.*

**Hypothesis 2b (H2b).** *Employees' emotional labour is negatively related to job satisfaction.*

### 2.3. EL as a Mediator

Referring to the role theory narrative adopted by researchers to describe FLEs' attitudes and behaviours (e.g., [84]), role expectations involve the FLEs' obligation to persistently perform their duties as indicated in their job descriptions [53]. Moreover, according to the emotion regulation theory, these employees are obliged to comply with the identified rules corresponding to their roles and to manage their emotions accordingly. Under these circumstances, employees will mobilise EL to adhere to standardised role expectations and fulfil the needed operational and emotional requirements of their jobs.

Studies claim that FLEs often need to engage in demanding encounters with customers [85]. Since interactions in hospitality settings can be very emotionally charged [29], FLEs are hence subject to EE [86]. Chen et al. [87] point out that a main feature of hospitality work that causes stress is the amount of EL performed by workers. Research also concludes that a rise in EE is experienced when FLEs are expected to participate in EL on an ongoing basis [77,80,88]. Additionally, job standardisation has a positive effect on the JS of FLEs [19]. However, FLEs may be required to comply with standards of display rules [31]. These rules require that FLEs regulate their emotions, thus engage in EL. Therefore, FLEs are increasingly subject to emotional dissonance [76], which negatively affects JS [45].

In addition to the aforesaid effects of EL on employees' EE and JS, these proposed relations signify the mediating role that EL may play between standardisation and employee EE and JS. Assessing this possible mediating role is considered since EL is identified by Grandey [78] as an occupational requirement and by Koc [89] as a necessity to improve hospitality employees' effective functioning. Furthermore, Hochschild [31] states that the emotional style of a service is part of the service itself.

Considering the abovementioned propositions, this paper suggests that EL is both a job outcome and a job requirement. It also proposes that EL mediates the relation between standardisation and EE and JS, as stated in the following hypotheses:

**Hypothesis 3a (H3a).** *Emotional labour fully mediates the relation between standardisation and emotional exhaustion.*

**Hypothesis 3b (H3b).** *Emotional labour fully mediates the relation between standardisation and job satisfaction.*

### 2.4. Conceptual Model

In sum, this study suggests that standardisation is negatively related to FLEs' EL and that EL is positively related to EE and negatively related to JS. Moreover, EL is believed to fully mediate the relation between standardisation and EE and between standardisation and JS. These hypothesised relations are presented in the conceptual model (see Figure 1).

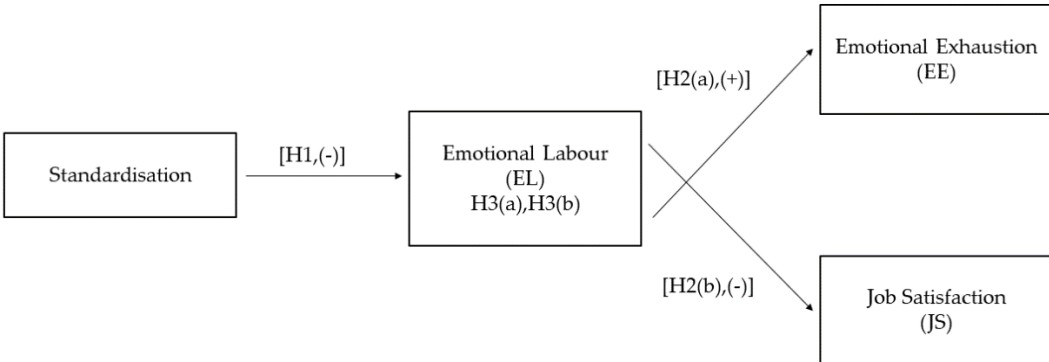

**Figure 1.** Conceptual Model.

## 3. Methodology

### 3.1. Measures

All items were adopted from relevant empirical research. To assess the standardisation variable, this study used Hsieh and Hsieh's [16] job standardisation scale. Chu and Murrmann's [90] Hospitality Emotional Labour Scale (HELS) was utilised to assess EL. JS was measured using three items from the job satisfaction subscale of the Michigan Organizational Assessment Questionnaire (MOAQ) developed by Cammann et al. [91]. Finally, EE was assessed using Maslach and Jackson's [92] Burnout Inventory (MBI)–emotional exhaustion subscale. Demographic information was also gathered to determine the study participants' profiles.

### 3.2. Method

A purposive/judgemental sampling technique was applied. FLEs working in restaurant operations in Lebanon were recruited. The constructed questionnaire included 32 items. Applying Westland's [93] rule-of-thumb of five, where the sample size is based on five times the number of variables, and Kerlinger and Lee's [94] method, where the study sample for factor analysis must be at least 5–10 times the number of survey questions, the minimum number of respondents was calculated to be 160.

The adopted questionnaire was translated to Arabic and then back-translated [95] before being revised and proofread by a certified translator to ensure compatibility. Emails requesting permissions to conduct the survey were sent to human resources or operations managers of selected operations. The email stated that confidentiality was guaranteed and that no internal information was required. Of the 29 restaurants and hotel food and beverage outlets contacted, 13 rejections and 16 permissions were received. The number of FLEs working in these operations totalled 789. Fifteen employees participated in the pilot study to ensure that the items were understandable.

The data were collected over a four-week period from July to August 2020. The respondents could choose either the English or the Arabic language. On the cover page, the respondents were informed that the survey would take 10 min to complete and that they were not expected to include their own or their employer's information. A Google form link was sent to the human resources departments, followed by three reminders sent at three-day intervals. By the end of the period, 292 complete replies were received, yielding a 37.01% response rate. The 292 completed questionnaires were all accepted due to Google form's built-in requirement for respondents to answer all questions so that the form can be finalised and submitted.

## 4. Data Analysis

### 4.1. Respondents' Profiles

The overall sample characteristics are shown in Table 1. Because of their significance in shaping and understanding the respondents' attitudes, some highlights are worth emphasising. For instance, most of the sampled FLEs in this study were young adults, with a cumulative 86.3% within the 18–34 age range, and 79.5% had a bachelor's or a master's degree. Furthermore, 42.5% (the largest segment of the sample) held entry-level positions, thus performing basic operating functions. Eighty respondents (27.4%) had worked for the same employer for one to two years, while 91.4% of the respondents had a standard full-time job.

**Table 1.** Respondents' profiles (*n* = 292).

| Section | | Sample | | |
|---|---|---|---|---|
| | | Frequency | Percentage (%) | Cumulative Percentage (%) |
| Gender | Male | 183 | 62.7 | 62.7 |
| | Female | 109 | 37.3 | 100.0 |
| | Cumulative | 292 | 100% | |
| Age | 18–24 | 124 | 42.5 | 42.5 |
| | 25–34 | 128 | 43.8 | 86.3 |
| | 35–44 | 35 | 12.0 | 98.3 |
| | 45–54 | 5 | 1.7 | 100.0 |
| | Cumulative | 292 | 100% | |
| Marital status | Single | 209 | 71.6 | 71.6 |
| | Engaged | 19 | 6.5 | 78.1 |
| | Married | 56 | 19.2 | 97.3 |
| | Separated | 2 | 0.7 | 97.9 |
| | Divorced | 6 | 2.1 | 100.0 |
| | Cumulative | 292 | 100% | |
| Education level | Intermediate education | 8 | 2.7 | 2.7 |
| | High school graduate | 43 | 14.7 | 17.5 |
| | Bachelor's degree | 185 | 63.4 | 80.8 |
| | Master's degree | 47 | 16.1 | 96.9 |
| | PhD | 1 | 0.3 | 97.3 |
| | Other (vocational education) | 8 | 2.7 | 100.0 |
| | Cumulative | 292 | 100% | |
| Position | Trainee | 16 | 5.5 | 5.5 |
| | Entry-level agent | 124 | 42.5 | 47.9 |
| | Supervisor | 69 | 23.6 | 71.6 |
| | Manager | 83 | 28.4 | 100.0 |
| | Cumulative | 292 | 100% | |
| Tenure | Less than 1 year | 35 | 12.0 | 12.0 |
| | 1 and less than 2 years | 80 | 27.4 | 39.4 |
| | 2 and less than 4 years | 67 | 22.9 | 62.3 |
| | 4 and less than 6 years | 43 | 14.7 | 77.1 |
| | 6 years or more | 67 | 22.9 | 100.0 |
| | Cumulative | 292 | 100% | |
| Monthly salary * | Less than 750,000 LL | 46 | 15.8 | 15.8 |
| | 750,001–1,500,000 LL | 148 | 50.7 | 66.4 |
| | 1,500,001–2,250,000 LL | 47 | 16.1 | 82.5 |
| | 2,250,001–3,000,000 LL | 20 | 6.8 | 89.4 |
| | More than 3,000,000 LL | 31 | 10.6 | 100.0 |
| | Cumulative | 292 | 100% | |
| Type of employment | Part-time | 25 | 8.6 | 8.6 |
| | Full time | 267 | 91.4 | 100.0 |
| | Cumulative | 292 | 100% | |

* LL = Lebanese Liras.

### 4.2. Data Analysis

Anderson and Gerbing's [96] recommended two-step approach was applied to analyse the data. The preliminary step comprised confirmatory factor analysis [96,97] and composite reliability [98]. The chi-square of the full model was compared with the chi-square of the partial one in the second step. To evaluate the hypothesised relations, structural equation modelling (SEM) was employed using LISREL 8.30 [99].

The results of the exploratory factor analysis showed no evidence of multicollinearity. The results of the confirmatory factor analysis pinpointed the need for the omission of

several items (i.e., nonsignificant t-values and standardised loadings below 0.50). Hence, three items from EL and two items from EE were discarded from further analysis.

The final results confirmed an acceptable fit of the factor measurement model: $\chi^2 = 839.95$, df = 672, $\chi^2/\text{df} = 1.249$, Normed Fit Index (NFI) = 0.96, Comparative Fit Index (CFI) = 0.97, Incremental Fit Index (IFI) = 0.89, Root-Mean square Residual (RMR) = 0.28, Goodness-of-Fit Index (GFI) = 0.81, Adjusted GFI (AGFI) = 0.79, Parsimony GFI (PGFI) = 0.74, Root Mean Square Error of Approximation (RMSEA) = 0.045. The results showed that all observable indicators significantly loaded on their respective constructs. The standardised loadings ranged from 0.78 to 0.94. The average variance extracted by each latent variable was above 0.50 for all constructs (standardisation = 0.76, EL = 0.75, EE = 0.88, and JS = 0.75), which confirmed convergent validity [97]. Likewise, the composite reliability of each construct (standardisation = 0.89, EL = 0.93, EE = 0.91, and JS = 0.91) was above the accepted cut-off level. All the mentioned information is presented in Table 2. In order to determine whether common method variance was a concern in this study, Harman single-factor test was applied. That is to say, all items related to standardization, EL, EE, and JS were loaded on a single factor through exploratory factor analysis. The single factor merely explained 20% of the variance. Conferring to this result, common method variance was not a serious concern.

**Table 2.** Confirmatory factor analysis results.

| Items | Standardised Loading | *t*-Value | AVE | CR |
|---|---|---|---|---|
| *Standardisation* | | | 0.73 | 0.89 |
| 1.   There are no standard operating procedures in this company. | 0.85 | 13.15 | | |
| 2.   We have to follow strict operating procedures at all times. | 0.85 | 14.62 | | |
| 3.   Whatever situation arises, we have procedures to follow in dealing with it. | 0.80 | 14.53 | | |
| 4.   Our company effectively uses automation to achieve consistency in serving customers. | 0.88 | 14.16 | | |
| 5.   Everyone has specific operating procedures to follow. | 0.88 | 15.67 | | |
| *Emotional Labour* | | | 0.75 | 0.93 |
| 1.   I put on a "mask" in order to express the right emotions for my job. | 0.88 | 15.30 | | |
| 2.   I have to cover up my true feelings when dealing with customers. | 0.86 | 14.43 | | |
| 3.   When dealing with customers, I display emotions that I am not actually feeling. | 0.85 | 15.40 | | |
| 4.   I fake the emotions I show when dealing with customers. | 0.88 | 15.15 | | |
| 5.   My smile at customers is often not sincere. | 0.79 | 15.34 | | |
| 6.   My interactions with customers are very robotic. | - | - | | |
| 7.   I put on an act in order to deal with customers in an appropriate way. | 0.79 | 14.29 | | |
| 8.   When dealing with customers, I behave in a way that differs from how I really feel. | 0.85 | 14.93 | | |
| 9.   I fake a good mood when interacting with customers. | 0.78 | 14.39 | | |
| 10.   I try to talk myself out of feeling what I really feel when helping customers. | - | - | | |
| 11.   I have to concentrate more on my behaviour when I display an emotion that I don't actually feel. | 0.85 | 15.75 | | |
| 12.   I try to actually experience the emotions that I must show when interacting with customers. | 0.94 | 16.27 | | |
| 13.   I try to change my actual feelings to match those that I must express to customers. | 0.91 | 17.10 | | |

**Table 2.** *Cont.*

| Items | Standardised Loading | *t*-Value | AVE | CR |
|---|:---:|:---:|:---:|:---:|
| 14.　I work at calling up the feelings I need to show to customers. | 0.89 | 16.45 | | |
| 15.　When dealing with customers, I attempt to create certain emotions in myself that present the image that my company desires. | - | - | | |
| *Emotional Exhaustion* | | | 0.85 | 0.91 |
| 1.　I feel emotionally drained from my work. | 0.89 | 17.05 | | |
| 2.　I feel used up at the end of the workday. | - | - | | |
| 3.　I feel fatigued when I get up in the morning and have to face another day on the job. | 0.93 | 17.71 | | |
| 4.　Working with people all day is really a strain for me. | 0.94 | 17.27 | | |
| 5.　I feel burned out from my work. | 0.89 | 16.98 | | |
| 6.　I feel frustrated with my job. | 0.90 | 17.10 | | |
| 7.　I feel that I'm working too hard on my job. | - | - | | |
| 8.　Working with people directly puts too much stress on me. | 0.94 | 17.23 | | |
| 9.　I feel like I'm at the end of my rope. | 0.94 | 17.35 | | |
| *Job Satisfaction* | | | 0.75 | 0.91 |
| 1.　All in all, I am satisfied with my job. | 0.89 | 15.63 | | |
| 2.　In general, I don't like my job. (reversed) | 0.86 | 14.87 | | |
| 3.　In general, I like working here. | 0.88 | 15.45 | | |

$\chi^2$ = 839.95, df = 672, $\chi^2$/df = 1.249, NFI = 0.96, CFI = 0.97, IFI = 0.89, RMR = 0.28, GFI = 0.81, AGFI = 0.79, PGFI = 0.74, RMSEA = 0.045.
Notes: All loadings are significant at the 0.01 level. (-) Dropped during confirmatory factor analysis. AVE = average variance extracted, CR = composite reliability.

The coefficient alpha values were also greater than 0.70. Therefore, all measures were reliable [98]. Table 3 provides information regarding the means, standard deviations, and correlations of the observed variables. The study applied a two-tailed test for the assessment of the correlations.

**Table 3.** Means, standard deviations, and correlations of observed variables.

| Variables | 1 | 2 | 3 | 4 |
|---|:---:|:---:|:---:|:---:|
| 1.　Standardisation | - | | | |
| 2.　Emotional Labour | −0.258 ** | - | | |
| 3.　Emotional Exhaustion | −0.253 * | 0.252 * | - | |
| 4.　Job Satisfaction | 0.280 * | −0.268 ** | −0.289 * | 0.337 ** |
| **Mean** | 4.29 | 4.34 | 3.63 | 3.64 |
| Standard deviation | 1.81 | 1.87 | 1.39 | 1.38 |
| Cronbach's alpha | 0.92 | 0.93 | 0.96 | 0.89 |

Note: The composite score was computed for each variable. * $p < 0.05$, ** $p < 0.01$ (two-tailed test).

### 4.3. Hypotheses Testing

As stated previously, the second step consisted of testing the hypotheses through SEM [99]. As shown in Table 4, the hypothesised model fitted the data well ($\chi^2$ = 306.27, df = 115, $\chi^2$/df = 2.66, NFI = 0.96, CFI = 0.98, IFI = 0.98, GFI = 0.92, AGFI = 0.86, PGFI = 0.72, RMSEA = 0.079) when compared with the alternative model. The results demonstrated a significant relation between standardisation and EL ($\beta_{21}$ = −0.66, $t$ = −3.93). Hence, H1 is supported. EL is also significantly related to EE ($\beta_{32}$ = 0.62, $t$ = 4.14) and JS ($\beta_{42}$ = −0.59, $t$ = −3.85). Thus, H2a and H2b are supported as well. To support the mediation hypotheses, H3a and H3b, the Sobel test results determined that EL fully mediated the relation

between standardisation and EE (z-score = −3.14) and between standardisation and JS (z-score = −2.76), respectively. To conclude, the results supported all hypotheses (see Figure 2).

**Table 4.** Results of model comparison (full mediation and partial mediation).

|  | $\chi^2$ | df | $\Delta\chi^2$ | $\Delta$df |
|---|---|---|---|---|
| **1. Hypothesised model (Full mediation)** | 306.27 | 115 | - | - |
| Standardisation → Emotional Labour | | | | |
| Emotional Labour → Emotional Exhaustion, Job Satisfaction | | | | |
| **2. Alternative model I (Partial mediation)** | 303.11 | 112 | 3.16 | 3 |
| Standardisation → Emotional Labour | | | | |
| Standardisation → Emotional Labour, Job Satisfaction | | | | |
| Emotional Labour → Emotional Exhaustion, Job Satisfaction | | | | |
| **3. Alternative model** | 302.62 | 111 | 3.65 | 4 |
| Standardisation → Emotional Labour | | | | |
| Emotional Labour → Emotional Exhaustion | | | | |
| Emotional Exhaustion → Job Satisfaction | | | | |

Note: Model fit statistics: $\chi^2$ = 306.27, df = 115, $\chi^2$/df = 2.66, NFI = 0.96, CFI = 0.98, IFI = 0.98, GFI = 0.92, AGFI = 0.86, PGFI = 0.72. RMSEA = 0.079. The hypothesised model appears to yield a better fit to the data than the alternative models.

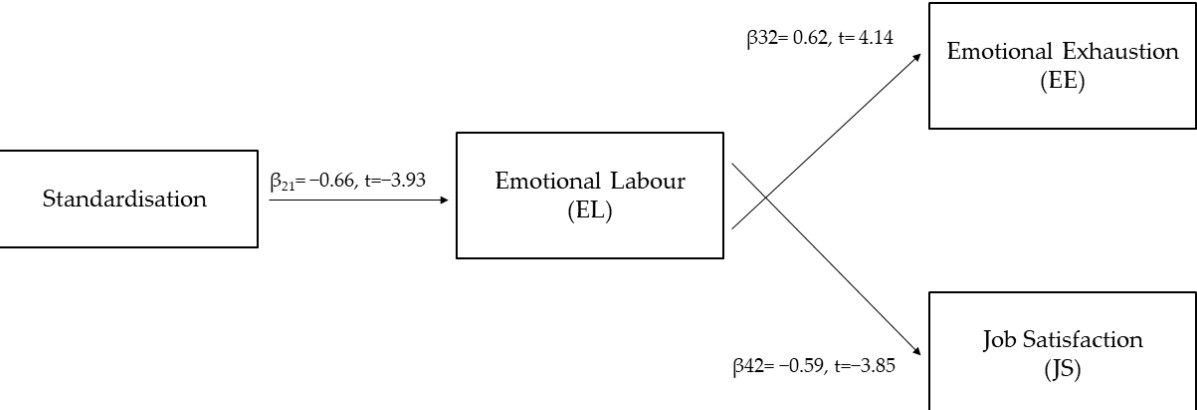

**Figure 2.** Hypotheses testing results using SEM.

## 5. Discussions, Theoretical, and Practical Implications

### 5.1. Discussion

This study tested the links between the standardisation of FLEs' tasks in Lebanese restaurants and employees' EL, the relationship between EL and EE and JS, and the mediating role of EL in the relationships between standardisation from one side and EE and JS from the other side. This study examined the data collected from 292 FLEs. The following paragraphs summarise the empirical findings of this study.

First, the results suggested that the standardisation of FLEs' jobs has a negative relationship with EL. That is, when FLEs follow standards to complete their daily tasks, they become less emotionally involved in their interactions with customers. These results are consistent with Morris and Feldman's [76] that the more psychological energy and physical commitment the service jobs demand from employees, the more emotional displays these will involve. Further, this finding confirms the study proposition that the degree to which FLEs in hospitality operations regulate their emotions is relative to the degree of

standardised rules, policies, and regulations imposed in the workplace. The findings also fill a research gap in the relationship between standardisation and emotional labour.

Second, the findings indicated that engagement in EL might trigger EE. In other words, when FLEs increasingly mobilise their emotional efforts and are increasingly emotionally involved in their daily interactions with customers, they are more likely to be emotionally exhausted. This finding is in accordance with various studies (i.e., [37,45,78–80]) which proposed that service providers experience greater levels of EE because their emotions are constantly either suppressed or exaggerated.

Third, the findings indicated that engagement in EL might cause reduced employee JS. That is, when FLEs put high emotional efforts in the workplace, they tend to be less satisfied with their jobs. This finding is consistent with those of previous studies [32,45], which considered that EL causes emotional dissonance with job dissatisfaction as one of its outcomes.

Moreover, the findings revealed that EL mediates the relationships between standardisation, and FLEs' EE and JS. In other words, when FLEs function in a highly standardised work environment, they are less likely to put EL in their daily interactions with customers, which in turn leads to a decreased level of EE and a favourable level of JS. Else, in the absence of job standards, FLEs are expected to increasingly engage in EL, they will then experience a higher level of EE and a decreased level of JS. Thus, EL is a critical significant mechanism that explains the links between standardisation and FLEs' EE and JS. This finding coincides with previous propositions (i.e., [31,78]) that considered EL as part of the service itself and as an occupational requirement.

*5.2. Theoretical Implications*

This study expands the use of the role theory and the emotion regulation theory to examine the effects of job standardisation on EL and the mediating effect of EL on FLEs' EE and JS in the food service industry.

First, the study contributes to the literature on standardisation and allows further assessment of standardisation in the workplace. Standardised customer service jobs adopt contact and control processes that managers introduce to handle service workers [100]. Managers then have these processes passed on to their FLEs to (1) increase service effectiveness by reducing the variance associated with task performance [101] and (2) meet customer expectations for fast and professional service delivery [13]. This study has examined the possible influences of standardisation on FLEs in service firms, where employees are emotional beings who are expected to operate differently from machines. The study concludes that standardisation has a negative effect on employee EL. Specifically, employees make less emotional effort when operations are highly standardised; thus, they are less expressive and emotionally involved in the service encounter.

Second, based on the emotion regulation theory, this study finds a significant effect of EL on job outcomes. This suggests that when FLEs use their emotional capacity to deal with job demands, they become emotionally exhausted and dissatisfied with their jobs.

Third, this research contributes to understanding EL as mediating the relation between standardisation and employee EE and JS in food service operations. Since EL is a process of managing effect and affective expressions at work [31], service companies require employees to control their emotions in service encounters to improve customer satisfaction [102]. EL research in the hospitality industry has primarily concentrated on favourable performance [103]. The present study has dealt with EL as (a) a requirement for service operations and (b) a mediator between standardisation and employee EE and JS. The findings show that EL plays a negative role in food service operations. It has been proven that EL is negatively related to JS and positively related to EE. FLEs are expected to use EL even when their job is routinised, and they work in a continuous loop of incidents. This is because service encounters can be so emotionally charged [29]. Therefore, job standardisation is expected to remain a tool for decreasing FLE performance variance [101]. However, standardising tasks will not eliminate the true essence of a service job, which

is highly driven; thus, it involves psychological and emotional triggers and reactions. Employees should have the ability to react genuinely to guests. This study concludes that EL is a natural tool utilised by employees when dealing with customers, but a degree of autonomy should be allowed to normalise the interactions with guests so that FLEs are neither emotionally exhausted nor dissatisfied with their job role.

*5.3. Practical Implications*

With reference to the emotion regulation theory, Yin et al. [77] argue that when people control their emotions, their bodies turn resources into energy to react to the current scenario, leaving less energy available for other duties. This resource drain makes the control of emotions harmful to the autonomous nervous system. In the long term, such resource demands can lead to physical or psychological illness, including EE. Moreover, Grandey [45] argues that EL may lead to dissonance, which may eventually cause employee dissatisfaction. A thorough understanding of the effects of EL will provide food service managers and operators with insights to enhance JS and decrease EE through multiple mechanisms proposed hereafter.

Managers and operators should pay close attention to the extent to which their FLEs put on an EL act daily. Because of their role requirements, if FLEs invest a high degree of EL, practitioners should expect high levels of employee dissatisfaction and EE, which are primary ingredients of burnout [104]. In maintaining a high level of service quality, employees are at the brink of emotional depletion [80]. They will have a higher intention to leave and transfer to a less emotionally demanding job. Therefore, managers should impose policies that take into consideration the psychological consequences of FLEs service tasks. Furthermore, planning jobs and positions should consider the emotional state that employees may reach because of the high level of standards imposed on them and the increased level of EL required to ensure that they serve customers according to such standards. In short, designing jobs in a way that the set of standards remain an element of the service process while leaving a recognisable space for genuine interaction will enable leaders of restaurant service teams to ensure employee JS and decreased levels of employee EE.

Managers should train employees to use their emotional capacity effectively and to harness their emotions in a way that will relieve the pressure caused by possible emotionally charged encounters. Organisational support can be presented as timely coaching of employees to express their true feelings when serving customers, learning techniques to balance emotions at work, and team meetings to share the best ways to deal with emotional job demands. Furthermore, practitioners can introduce on-the-job and off-the-job activities where employees may blow off steam and ensure that their emotional capacities are not depleted. These mechanisms should ensure effective employee advancement and retention of well-trained high-performing employees.

Finally, since EL is now one of the most critical work demands of today's dynamic service industry [105], this paper proposes the introduction of relevant course materials in the curricula of institutes that offer hospitality management programmes. Exposing future employees to the emotional requirements of customer-contact positions beforehand will prepare them for their prospective roles. Furthermore, providing technical training on how to apply various EL strategies in daily encounters will instruct future employees on the importance of EL and the best way to put it in practice.

## 6. Limitations and Future Research Directions

Several limitations of this study are worth mentioning. First, some data collection biases may be present because of the limited time and the specific location of the data collection. At that time, Lebanon was undergoing a severe financial crisis, which affected the schedule and payment of a considerable percentage of employees in food service operations. This might have affected their responses. Second, the results of the current study are based on cross-sectional data, which could highlight the concerns regarding

generalizability. Therefore, conducting a longitudinal study is truly encouraged. Third, the study did not include the role of customers, who comprise the other side of the encounters in food service operations. Future studies should assess the moderator effect of customers. Fourth, according to the role theory narrative, Fisk [106] breaks down ongoing service encounters into preconsumption, consumption, and postconsumption stages. Assessing the degree of emotional involvement at each stage might also be considered. Lastly, standardisation was assessed as a general work condition, with no specification of what job element of FLEs was standardised. Therefore, specific standardised elements (e.g., verbal service scripts, telephone encounters, etc.) could be considered for inclusion in future research.

**Author Contributions:** Conceptualization, S.I. and O.C.; methodology, O.C. and M.B.; software, M.B.; validation, S.I., M.B. and O.C.; formal analysis, M.B.; investigation, S.I. and O.C.; data curation, O.C.; writing—original draft preparation, O.C. and S.I.; writing—review and editing, O.C., S.I. and M.B.; supervision, S.I. All authors have read and agreed to the published version of the manuscript.

**Funding:** This research received no external funding.

**Data Availability Statement:** The data presented in this study are available on request from the corresponding author. The data are not publicly available due to restrictions of privacy.

**Conflicts of Interest:** The authors declare no conflict of interest.

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
