# Peer review of "Impacts of Job Standardisation on Restaurant Frontline Employees: Mediating Effect of Emotional Labour"

_sustainability, doi:10.3390/su13031525_

Round 1

Reviewer 1 Report

In general, I felt the topic is very interesting and found it very timely. I really enjoyed reading this manuscript. This paper has several strengths, including: 1) Overall this is a well-structured and written paper, especially in the literature review, methods, and results sections; and 2) The introduction section is stated clearly, and hypotheses are addressed properly.

I believe the author(s) have done so much work for this manuscript, but there are a few concerns about the overall quality of the work. I am trying to provide my comments and observations and hope the author(s) find them helpful.

  1. The introduction section should include a last paragraph showing the structure of the paper.
  2. The introduction is good in terms of appropriateness of context and the purpose of study, although a brief overview of the hospitality industry would be welcome. In this sense, the following articles could be useful for you:
    • Ariza-Montes, A., Arjona-Fuentes, J. M., Han, H., & Law, R. (2018). Work environment and well-being of different occupational groups in hospitality: Job Demand–Control–Support model. International Journal of Hospitality Management, 73, 1-11.
    • Ariza-Montes, A., Hernández-Perlines, F., Han, H., & Law, R. (2019). Human dimension of the hospitality industry: Working conditions and psychological well-being among European servers. Journal of Hospitality and Tourism Management, 41, 138-147.
    • Hofmann, V., & Stokburger-Sauer, N. E. (2017). The impact of emotional labor on employees’ work-life balance perception and commitment: A study in the hospitality industry. International Journal of Hospitality Management, 65, 47-58.
    • Murphy, K., Torres, E., Ingram, W., & Hutchinson, J. (2018). A review of high performance work practices (HPWPs) literature and recommendations for future research in the hospitality industry. International Journal of Contemporary Hospitality Management.

  1. The author(s) should strengthen the role of EL as a mediator in the relation between standardisation and emotional exhaustion, on the one hand, and between standardisation and job satisfaction, on the other hand.
  2. The conceptual model should be the last subsection of Section 2. Literature review and hypotheses, in no case as a subsection of the methodology section.
  3. The author(s) should indicate in the conceptual model the direction (+) or (-) of the hypotheses.
  4. In the 3.2. Measures Section, the author(s) should include some examples of items for each scale (or he/she would direct the readers to Table 2, where they can see the items that compose the questionnaire).
  5. The author(s) should include a footnote at the end of Table 1, indicating the meaning of LL in the monthly salary variable (¿Lebanese Lira?).
  6. My main concern is the absence of a Discussion section. This is a problem that needs to be corrected. There was a missed opportunity to focus the discussion on what is new from this paper compared to the existing literature in this topic. The author(s) must elaborate a discussion section and tie their findings back to the research cited in the literature review and theory development section, and discuss how those findings confirm, alter or refute previous research on the subject. This will further highlight the unique contribution of your work above and beyond what is already known in the literature.

I hope you find my comments helpful and I wish the author(s) the best of luck in developing this research.

Author Response

Dear Reviewer;

Thanks a lot for your careful evaluation, comments, and recommendations. We re-edit our paper based on your comments and suggestions. We believe that our manuscript has improved significantly relative to initial submission with the help of the review process.

Our responses to the Reviewers’ comments and suggestions are shown below. Also, in the revised manuscript, we made the necessary adjustments. 

Sincerely,

Reviewer 2 Report

Dear authors,

I appreciate having the opportunity to review the manuscript entitled “Impacts of job standardisation on restaurant frontline employees: Mediating effect of emotional labour” (sustainability-1070340). I really enjoy your paper.

This research investigates the mediating effect of emotional labour on the association between job standardisation, and both emotional exhaustion and job satisfaction. The results demonstrated that job standardisation influences emotional labour negatively and that emotional labour positively affects emotional exhaustion and negatively influence job satisfaction. Moreover, emotional labour mediated the association between job standardisation and emotional exhaustion and job satisfaction.

  I believe that the current manuscript is adequate to be published at Sustainability, based on its appropriate capturing research questions, transforming those into hypotheses, testing the hypotheses, and presenting those clearly. However, I want to suggest some additional things to contribute to improving your paper.

  [1] Analytical Strategy

- The overall structure of your research is based on cross sectional data. Thus, I think that you should provide adequate explanations about the limitation.

- It is not easy for psychologists to accept and trust the result of indirect effect test from cross-sectional data. I think that you had better try to alleviate the issue pertinent to the impact of third variables or alternative explanations [1].

[1] Conway, J. M.; Lance, C. E. What reviewers should expect from authors regarding common method bias in organizational research. J. Bus. Psychol. 2010, 25, 325-334.

- In addition, the authors only conducted a CFA to demonstrate the measurement model is adequate. Despite of its efforts to achieve it, it is not enough since this paper omitted the model comparisons among all alternative models including 4 factor model, 3 factor model, 2 factor model, and single factor model. I recommend the authors to conduct the model comparison tests by utilizing chi-square difference tests.

 I wish these comment may help you to improve your paper. Good luck.

Author Response

Dear Reviewer;

Thanks a lot for your careful evaluation, comments, and recommendations. We re-edit our paper based on your comments and suggestions. We believe that our manuscript has improved significantly relative to the initial submission with the help of the review process.

Our responses to the Reviewers’ comments and suggestions are shown below. Also, in the revised manuscript, we made the necessary adjustments. 

Sincerely,

Round 2

Reviewer 1 Report

Looks much better to me now and happy to accept it in its current version.

Reviewer 2 Report

 Dear authors, 

 Thank you for your efforts to revise the paper. The revision is enough.